# Mapping global inputs and impacts from of human sewage in coastal ecosystems

Cascade Tuholske [1,2,3☯]*, Benjamin S. Halpern[2,4☯], Gordon Blasco[2], Juan Carlos Villasenor[4], Melanie Frazier[2], Kelly Caylor[1,4]

**1** Department of Geography, University of California, Santa Barbara, CA, United States of America,
**2** National Center for Ecological Analysis and Synthesis, University of California, Santa Barbara, CA, United States of America, **3** Center for International for International Earth Science Information Network, the Columbia Climate School and its Earth Institute, Columbia University, Palisades, NY, United States of America, **4** Bren School of Environmental Science & Management, University of California, Santa Barbara, CA, United States of America

☯ These authors contributed equally to this work.
* cascade@ucsb.edu

**Data Availability Statement:** All data are available at Knowledge Network for Biocomplexity (KNB) https://github.com/OHI-Science/GlobalWasteWater.

## Abstract

Coastal marine ecosystems face a host of pressures from both offshore and land-based human activity. Research on terrestrial threats to coastal ecosystems has primarily focused on agricultural runoff, specifically showcasing how fertilizers and livestock waste create coastal eutrophication, harmful algae blooms, or hypoxic or anoxic zones. These impacts not only harm coastal species and ecosystems but also impact human health and economic activities. Few studies have assessed impacts of human wastewater on coastal ecosystems and community health. As such, we lack a comprehensive, fine-resolution, global assessment of human sewage inputs that captures both pathogens and nutrient flows to coastal waters and the potential impacts on coastal ecosystems. To address this gap, we use a new high-resolution geospatial model to measure and map nitrogen (N) and pathogen—fecal indicator organisms (FIO)—inputs from human sewage for ~135,000 watersheds globally. Because solutions depend on the source, we separate nitrogen and pathogen inputs from sewer, septic, and direct inputs. Our model indicates that wastewater adds 6.2Tg nitrogen into coastal waters, which is approximately 40% of total nitrogen from agriculture. Of total wastewater N, 63% (3.9Tg N) comes from sewered systems, 5% (0.3Tg N) from septic, and 32% (2.0Tg N) from direct input. We find that just 25 watersheds contribute nearly half of all wastewater N, but wastewater impacts most coastlines globally, with sewered, septic, and untreated wastewater inputs varying greatly across watersheds and by country. Importantly, model results find that 58% of coral and 88% of seagrass beds are exposed to wastewater N input. Across watersheds, N and FIO inputs are generally correlated. However, our model identifies important fine-grained spatial heterogeneity that highlight potential tradeoffs and synergies essential for management actions. Reducing impacts of nitrogen and pathogens on coastal ecosystems requires a greater focus on where wastewater inputs vary across the planet. Researchers and practitioners can also overlay these global, high resolution, wastewater input maps with maps describing the distribution of habitats and species, including humans, to determine the where the impacts of wastewater pressures are highest. This will

**Funding:** Support for BSH, JCV, CPT, MF, and GB was provided by National Philanthropic Trust. Additional funding for CPT came from the UC President's Dissertation Year Fellowship program and the Earth Institute Postdoctoral Fellowship Program, Columbia University. Partial support for KC was provided by National Science Foundation SES-1832393. Funding for computation infrastructure used by CT, JCV, GB and MF was provided by the National Center for Ecological Analysis and Synthesis.

**Competing interests:** The authors have declared that no competing interests exist.

help prioritize conservation efforts.Without such information, coastal ecosystems and the human communities that depend on them will remain imperiled.

## Introduction

Coastal marine ecosystems face myriad human pressures, including those from climate change, overfishing, offshore commercial uses, and land-based activities such as agriculture [1]. Land-based stressors, such as nutrient and chemical pollution run-off, link coastal marine systems to terrestrial human activities and represent important and often dominant stressors in coastal ecosystems [2–4]. They also amplify risks from other anthropogenic stressors such as climate change due to potential synergistic interactions among stressors [5, 6]. Agricultural fertilizers and livestock waste have long been recognized as major contributors to coastal eutrophication, harmful algal blooms, hypoxic zones, or anoxic dead zones [2, 4, 7, 8], with evidence that these impacts can lead to substantial habitat degradation or loss [9, 10], fisheries declines [11], and beach closures, among other environmental consequences. These impacts are likely to get worse with global climate change [5, 9], particularly affecting coastal ecosystems such as seagrasses [12], mangroves [13], salt marshes [14], and coral reefs [15] which are vulnerable to even modest levels of nutrient input [16, 17].

In contrast to agricultural inputs, few high-resolution studies have measured and mapped the impacts from human sewage on coastal ecosystems [16, 17]. Human sewage is both an human health and ecological health issue [18] because of its dual role in increasing coastal nitrogen (N) and transporting pathogens and other health hazards [18]. Its input is likely accelerating due to population growth and increasing dietary protein consumption not just in wealthy countries [19], but also in emerging economies like China and Brazil where diets are shifting toward heavy protein consumption [20, 21].

Managing wastewater, particularly for non-sewered systems, has been a cornerstone of public health intervention around the world [22, 23], and human sewage is known to contribute substantially to anthropogenic N inputs [24–28]. But the solution space for wastewater treatment can look very different for mitigating excess N input versus pathogen risk. Potential tradeoffs and synergies make it essential to coordinate assessment and management actions. For example, most sewered systems concentrate and emit N while removing most pathogens [29–31], while direct input exposes people to pathogens, but allows natural ecosystems to process much of the N and in some cases can mitigate pathogen exposure [32]. Yet research on nutrient input and pathogen risk from wastewater have largely occurred in isolation [33]. Despite clear risks to both coastal ecosystems and public health, we presently lack globally comparable, fine-scale estimates of both nutrient and pathogen inputs to coastal waters from human sewage across all coastal ecosystems and communities worldwide [33]. Given finite resources, without a fine-grained appraisal, policymakers presently have limited capacity to weigh options to protect coastal ecosystems and improve public health.

Existing models that estimate nitrogen inputs or pathogen inputs to surface water from human sewage suffer from limitations. Global nutrient river export models that account for human sewage rely on coarse-grained (0.5 by 0.5 degree) input data, focus primarily on the largest watersheds, are limited to assessing input from sewered systems [25–29, 34–38], and generally do not propagate inputs into coastal waters (instead focusing on watershed impacts). Yet, over 60% percent of the planet's population lack sewage connections (instead using open defecation, pit latrines, or septic tanks) [39], the vast majority (>99%) of the global coastline is

not adjacent to the mouth of the largest watersheds [3], and coastal ecosystems are known to be sensitive to N inputs.

Similarly, the handful of global pathogen surface water studies, which primarily estimate how human population growth and urbanization drive surface-water Cryptosporidium [40], suffer from drawbacks. Like global N models, global pathogen models rely on a common modeling design [40–43], are coarse-grained (0.5 by 0.5 degree), do not account for inputs from non-sewered populations, and do not map surface water flows to coastal ecosystems. Furthermore, while case studies can provide context for specific details–such as how fecal indicator bacteria respond to environmental conditions [44], the effect of future climate change and socioeconomic conditions [45] and continental-scale sensitivity and uncertainty analysis [46] global pathogen wastewater studies report results at national or larger scales. This crucially constrains our ability to assess the magnitude and distribution of coastal nutrient and pathogen stressors and develop regional and local policy recommendations and actions, much less compare fine-grained pathogen impacts to nutrient impacts on coastal communities and ecosystems.

Here, we present the first globally-comprehensive, fine-resolution (~1-km) assessment of N and pathogen—measured as fecal indicator organisms (FIO)—inputs from wastewater from nearly 135,000 watersheds and coastal areas that drain into the ocean. Our assessment leverages a high-resolution geospatial model to quantify and map the contribution of N from direct, septic, and sewer wastewater in 2015; to evaluate differences between N and pathogen inputs; and to estimate potential exposure of two key sensitive coastal habitats (coral reefs and seagrass beds) to these inputs. We estimate the total and relative contribution of N from different sources of sewage, and identify where opportunities for wastewater N mitigation exist, and how they may coincide or conflict with efforts to address pathogen inputs. Our work acknowledges the existing tradeoffs in managing N and pathogens for ecosystem and human health, but also identifies important synergies and win-win scenarios where interventions can produce benefits for humanity and the environment.

## Data & methods

### Overview of modeling approach

Global rasters of both nitrogen effluent (N, grams yr$^{-1}$) and pathogens as measured by FIO were developed following a similar approach. Population and settlement type data (~1km resolution) were combined with national level statistics on protein consumption (for nitrogen only) and the proportion of the population's access to various levels of wastewater treatment facilities. This included treated wastewater that drains to the ocean, as well as direct and septic effluent within 1km of coastlines or surface water. Once the global raster was completed, we summed total N and FIO for 142,625 watersheds or coastal areas, 134,846 of which flow into or are adjacent to the ocean (vs. internal draining). Watershed effluents were attributed to corresponding coastal pourpoints–the location where surface water 'pours' into the ocean. For N, at each pourpoint, the effluent was propagated into the coastal waters using a plume model based on a logarithmic decay function [1]. The effluent plumes were used to determine the extent to which different marine habitats are exposed to wastewater N. Finally, we benchmark our methods and findings throughout our analysis (S1 Text in S1 File).

We utilized Python v3.7, R v3.6.3, and GRASS GIS v7.8.3 programming languages to conduct these analyses. All data are available at Knowledge Network for Biocomplexity (KNB) https://doi.org/10.5063/F76B09 and code is available at https://github.com/OHI-Science/wastewater.

## Creating global rasters of marine-relevant FIO and nitrogen from wastewater

**Population & settlement type.** The driver of anthropological wastewater is human population; consequently, mapping wastewater inputs requires fine-grained population data. Within a country, access to treatment facilities depends on the level of urbanization (i.e., if the population is an urban or rural settlement) [47]. Our model used the European Global Human Settlement Population Layer version 4 (GHS-Pop), which provides population estimates for 2015 for raster cells at 30 arc-second resolution (1km at the equator). To identify rural and urban pixels [48], we used the 2015 Global settlement layer settlement model grid (GHS-SMOD) [48]. GHS-Pop allocates population from a census-derived, un-modeled gridded population dataset—the Gridded Population of the World version 4 (GPWv4)—to 1km grid cells based on built area estimations derived from Landsat imagery. GHS-SMOD further classifies GHS-Pop pixels into settlement typologies following a rural-to-urban continuum based on grid cell population size, density, and built area estimations. We aggregated the GHS-SMOD levels of settlement into two urban and rural classes based on the GHS-SMOD documentation in order to match sanitation and removal factors, as described below. While many gridded population datasets exist and show variation [49, 50], GHS-Pop is the only gridded population dataset that has corresponding settlement typologies (e.g. GHS-SMOD) developed from a common modeling paradigm.

**Sanitation facilities and removal rates.** Wastewater N and FIO inputs into the ecosystem from human sewage depend on the type of treatment. To estimate the amount of N removed from wastewater via sanitation facilities (or lack thereof), we used national statistics from WHO-UNICEF 2017 Joint Monitoring Project (JMP) [39], as has been done in previous studies [21, 27]. This dataset describes the proportion of the country's population with access to wastewater treatment in 2015. For most countries, data describing access to wastewater treatment were available for both urban and rural areas allowing us to account for the large within-country differences in access to sanitation. We group JMP wastewater treatment types into three broader categories: open defecation (direct), septic tanks (septic), or some level of sewered treatment (see S1 Table in S1 File for exact matching to JMP column).

We used access to sanitation to calculate N production coefficients. This can be thought of as the inverse of removal efficiency: higher numbers indicate more N reaches the ocean. Populations that use treatment types with lower removal rates will have a higher production coefficient.

We assigned the three sewage treatment categories of nitrogen removal efficiency based on literature [28]. We estimated treatment plants remove 55% of N [28], while septic removes 80% [51] and open defecation has no N removal. Therefore, the combined N removal factor of a given location is the sum of the products of the removal efficiency rates and percent of each type of access to sanitation. We acknowledge there is a significant range for these removal rates, based on landscape traits (e.g., soil type and vegetation cover) and especially for sewage treatment systems, which have N removal of 10% to 90% from primary to tertiary treatment systems [21, 27]. Because there is no global database of wastewater treatment facilities, our model used a uniform N removal rate for sewage systems. Nonetheless, as better data becomes available it will be possible to update the model results using the code that we made publicly available.

We modeled FIO production as a function of sanitation infrastructure and population size (similar to [40]) to evaluate the exposure of coastal ecosystems to FIOs. The amount of pathogens (FIO) removed from wastewater for the three input types was informed by a case study

that used more than 1,000 samples to determine FIO effluents across sanitation types [52]. These results for septic tanks have been confirmed elsewhere in the literature [53].

As noted above, most countries report access to treatment categories at the urban/rural level. When urban/rural data were not available, we used national-level statistics to impute N removal rates. For any jurisdictions not covered in the JMP dataset, we used the median of the UN geopolitical region [54] at development-level specific values (e.g. the median urban N removal rates of all countries in the same geopolitical region was used to impute the missing urban N removal rate). In the case of FIOs, for countries with missing data, we employed a random forest model to estimate FIO sanitation factors based on a nation's population size, the proportion of the population that is rural, and national-level FIO sanitation factors. In our analysis, random forests outperformed other commonly used algorithms, based on a 10-fold cross validation process for parameter tuning, followed by a 70/30 training/testing split for model selection (S1 Fig in S1 File). The final model was then used to predict missing FIO values. As with N factors, we used geopolitical medians to impute values for jurisdictions not covered in the JMP data. We used UN geopolitical medians [54] to impute the specific values.

Finally, for both FIO and N, our model assumed that populations with inputs from septic tanks or open defecation that are more than 1km away from surface waters and coastlines do not contribute to effluent totals. To accomplish this, we set the weight coefficient for septic tanks to 0 based on a 1km binary surface water raster, resampled from the 15 arc-second HydroSheds dataset [55].

**Quantifying amounts of effluent.** Following previous work [29], the model used FAO's Food Balance Sheets, which track national estimates for grams of protein consumed per capita per year, to calculate pixel-level N effluents [56]. For each pixel, we multiplied this value by the population size to obtain annual protein consumed in grams. Metabolic studies have established that 16% of the protein we ingest is excreted as different forms of N [57]. We thus scaled the pixel-level consumption by 0.16 to obtain N excretion base. These values were then multiplied by the N removal factors calculated above, which indicate the proportion of Nitrogen that escapes treatment and reaches the watersheds. For example, a value of 0.5 would indicate that 50% N excreted in a pixel would reach the pourpoint in a given watershed. This results in a 1km N (grams year$^{-1}$) effluent raster for 2015.

The FAO Balance Sheets contain information for 178 countries on per capita annual grams of protein consumed over a year. While this FAO dataset is the complete dataset on food consumption, it lacked information for 87 of the jurisdictions considered in our analyses, warranting some imputation. Previous literature has long-established a link between buying power and protein consumption [58]. Therefore, we constructed a database of national-level protein consumption and per capita GDP. A linear regression model (S2 Table and S2 Fig in S1 File) was used to impute protein consumption for places without per capita GDP data (n = 36).

For 51 jurisdictions, protein consumption data were not available. From these, 13 are uninhabited areas, and therefore protein consumption is not relevant. For the remaining 38 missing values, we calculated the median protein consumption for each UN geopolitical region and assigned this value to each country. S3 Table in S1 File shows the values and type of imputation used for each location. S3 Fig in S1 File shows the distribution and fit of imputed values, relative to observed known values. We were not able to obtain individual human production value estimates for FIOs and therefore we report FIO's in this analysis as a unitless measure that correlates with population.

**Watershed summation.** We used high resolution watershed vector data with corresponding pourpoints [1] (e.g. coastal location where watersheds empty into oceans). Boundaries for ~140,000 global basins were developed with an automated flow-accumulation process [59] on the basis of 30 arc-second (~ 1km) SRTM30 Digital Elevation Model (DEM) data [60, 61]. The

dispersion of the wastewater pollution into the ocean was then modeled from each basin's pourpoint which describes the location where freshwater enters the ocean. For each watershed with a pourpoint draining into the ocean (N = 134,846), we estimated total N, N by treatment type (sewered, septic, and open), and FIO inputs by summing the values for all grid cells falling within the watershed boundary. We then attributed each effluent type to the watershed's corresponding coastal pourpoint. To estimate each nation's discharge into the World's oceans, we summed the effluents for all the pourpoints falling within the country.

**Plumes & habitat overlap.** Pourpoint effluents were propagated with a coastal plume model at 1km resolution into the ocean using a decay function following [1]. We first removed pourpoints with < 1g N, retaining 66,309 coastal pourpoints. We then applied the plume model to distribute values into coastal waters based on a cost-path (in our case, 0.5% of the value in the previous cell) surface until a minimum threshold (0.05% of global maximum) is reached. This approach to modeling river plumes allows drivers to wrap around headlands and islands, but does not account for nearshore advection that acts to push N (or other inputs) in particular directions.

The modeled wastewater plume data was used to determine the exposure of coral and seagrass to N inputs from wastewater. We rasterized spatial polygon and point data describing global coral reef [62] and seagrass bed [63] locations to create raster maps of ~0.5 km resolution. These cells were then aggregated to ~1km resolution consistent with the output from the plume model. Cells including the habitat were classified as 1, and otherwise set to no value. We used a higher resolution for the initial rasterization to ensure a higher probability of capturing smaller polygon areas because the habitat is not identified in the cell unless it overlaps with the center of the raster cell. We extracted the N values for each sanitation system and the total N from all wastewater for each raster cell containing the habitat. Finally, we defined hotspots as habitat raster cells exposed to total wastewater N values equal to or greater than the 97.5th quantile determined across the entire range of the habitat.

## Results & discussion

Our model indicates that wastewater inputs into coastal waters account for an estimated 6.2Tg N of total anthropogenic N in coastal ecosystems globally. This estimate matches those from previous studies (wastewater N range: 4.0–7.2TgN), suggesting our model is reasonable, and is roughly 45% (40–52%) of the amount of N from agriculture [24, 27, 28, 64]. Of total wastewater N, 63% (3.9Tg N) comes from sewered systems, 5% (0.3Tg N) from septic, and 32% (2.0Tg N) from direct input. However, unlike previous studies, our fine-resolution model allows for identification of locations of particularly high (or low) inputs (Fig 1), and the relative contribution of different wastewater types to any location (S4a-S4d Fig in S1 File), highlighting substantial variation by country and region (Fig 2). The model suggests that wastewater input of N from watersheds into coastal waters is highly concentrated, with half (n = 67,308) of all watersheds adding no nitrogen or pathogens. Just 25 watersheds contribute approximately 46% (2.8Tg N) of global N inputs from wastewater into the ocean (S2a Fig and S4 Table in S1 File). These watersheds are concentrated in India, Korea and China but are also found in other continents, and a single watershed—the Chang Jiang (Yangtze) River in northern China—accounting for (11%) of global wastewater N.

The source of wastewater N varies across these watersheds from predominantly sewered (e.g., The Netherlands, USA, Argentina) to predominantly direct input (e.g., The Democratic Republic of the Congo, Nigeria, India, China) (Fig 2 and S4a-S4d Fig in S1 File). While previous studies have suggested that wastewater input of N is not significant in Africa and South America [25], these studies have not included septic and direct wastewater input. Our model

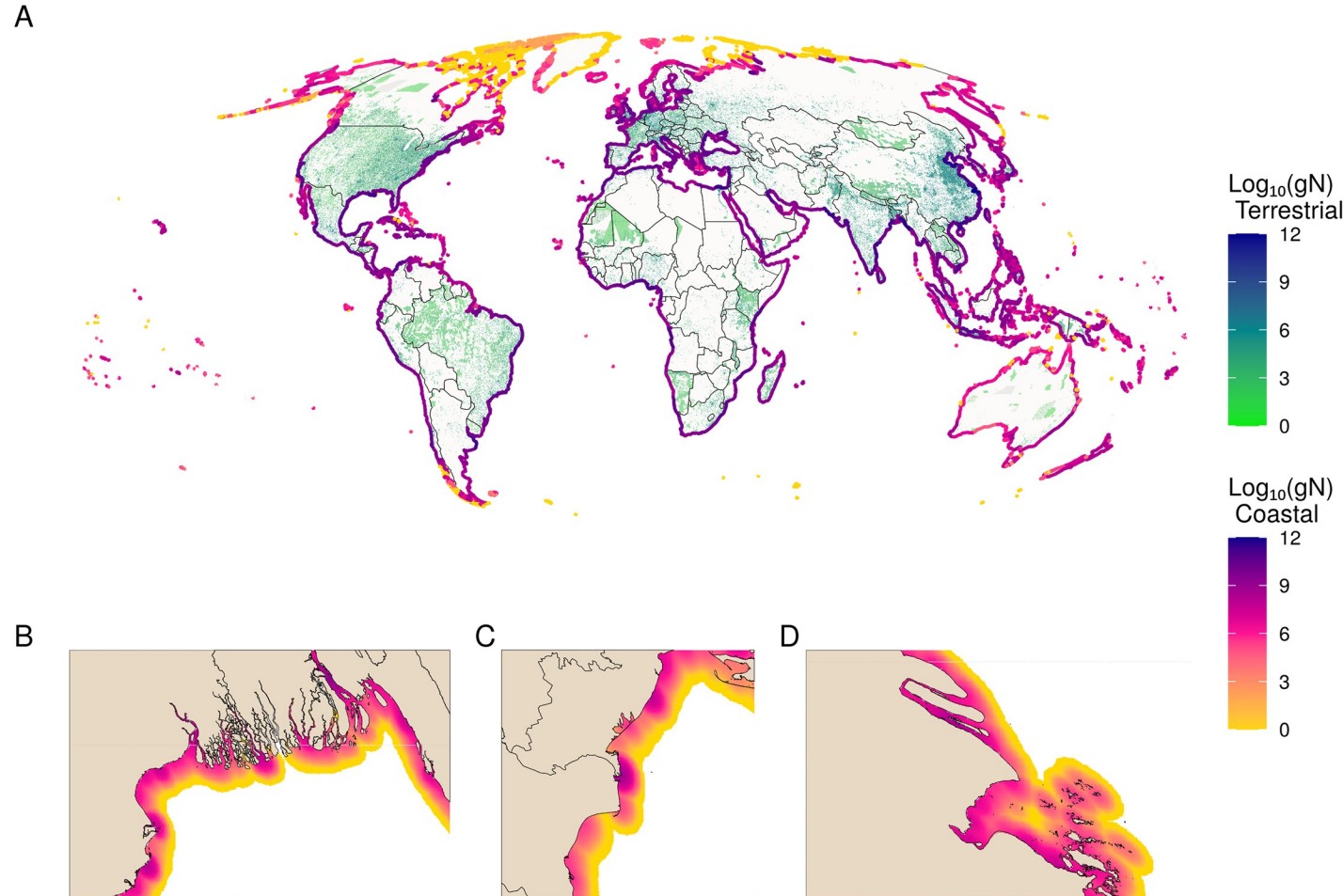

**Fig 1. Global distribution of total wastewater N.** A) Global map of the terrestrial sources (green to blue) and coastal diffusion of inputs (yellow to purple) of total wastewater N, measured in log10(gN) in both. Coastal plumes have been buffered to line segments to exaggerate patterns to be visible at the global scale. Insets show zoomed-in views of the B) Ganges, C) Danube, and D) Chang Jiang (Yangtze) Rivers, showing wastewater plumes at high resolution.

results indicate that wastewater input in these regions exceeds that of many countries in Europe and Asia (Fig 2). Furthermore, when normalized by watershed size, per-area N input is substantial for many more watersheds, with 'hotspots' in several Mediterranean countries, Mozambique, and the United States, among other places (S4 Table in S1 File), where the input is very high relative to the size of the watershed.

Concern for wastewater input of N in coastal habitats has focused on particularly vulnerable ones such as coral reefs and seagrass beds [15, 18]. Modeling the plume of wastewater N into coastal waters, we estimate about 58% of all coral reefs globally and 88% of all seagrass beds experience at least some anthropogenic N input from wastewater. We identify hotspots of exposure for coral occurring in China, Kenya, Haiti, India and Yemen, and hotspots of exposure for seagrass occurring in Ghana, Kuwait, India, Nigeria, and China (Fig 3). For 79% of coral reef areas with N inputs, the dominant source of wastewater N is from direct or septic input, and areas with particularly high inputs dominated by untreated wastewater include: East Africa (Kenya, Tanzania, Mozambique, Madagascar), Asia (Sri Lanka, Myanmar, Vietnam, Philippines), and, in the Caribbean, Haiti. For seagrass beds, 52% of areas with N input are dominated by direct or septic inputs, with Eastern Africa along the Gulf of Guinea (Benin,

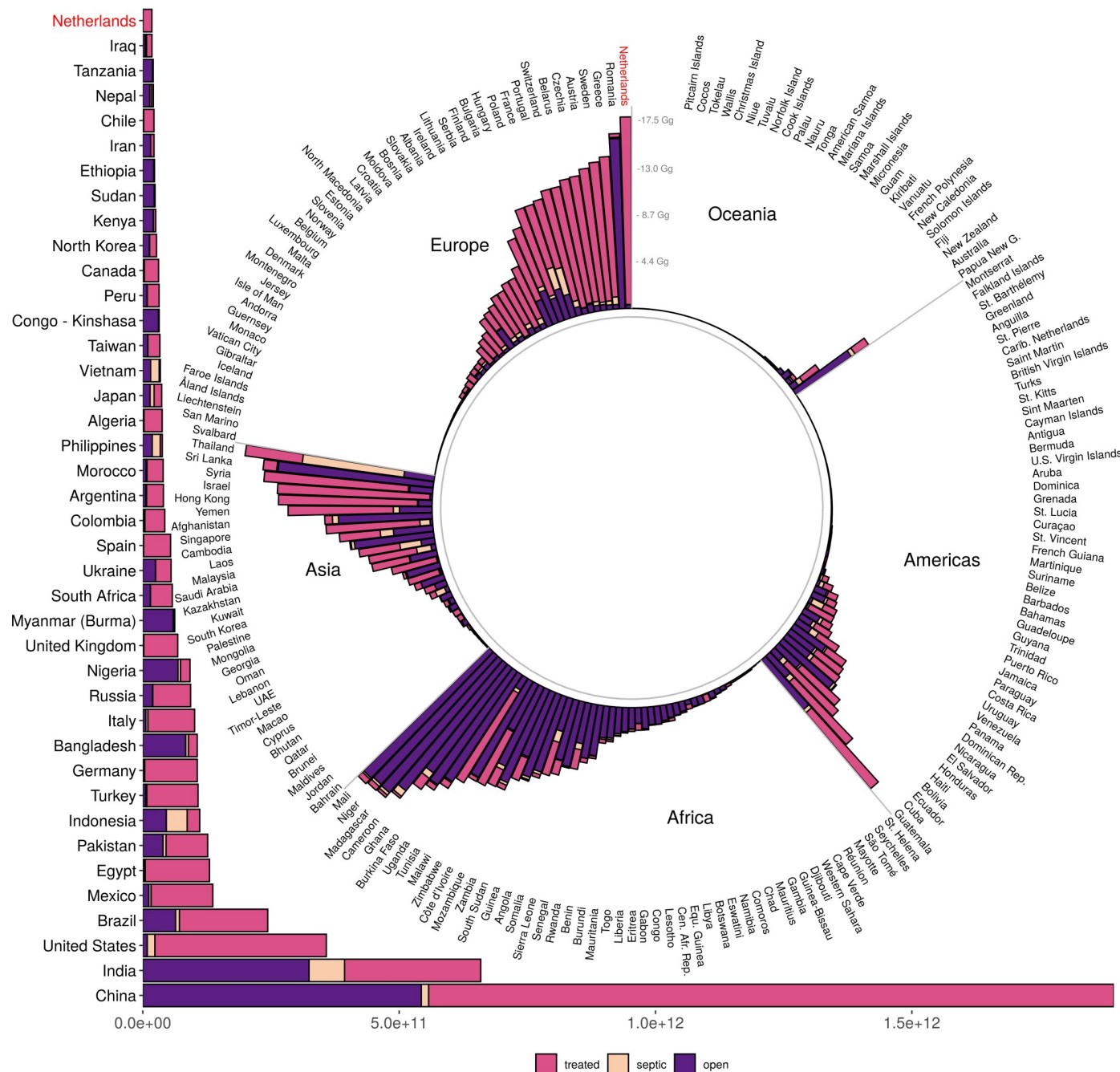

**Fig 2. Total nitrogen input into Exclusive Economic Zone (EEZ) waters of coastal countries, by source type (sewer, septic, direct).** The global total wastewater input is 6.2Tg N, with 3.9Tg from sewers, 0.3Tg from septic, and 2Tg from direct input. The top 40 countries are shown in the horizontal bar chart; remaining countries are in the pinwheel, grouped by continent or larger geographical region. Values for all countries are also reported in S5 Table in S1 File. Note that the Netherlands is shown in both places (in red) to help connect the scale of the two parts of the figure.

Nigeria, Togo, Ghana) having particularly high N inputs from untreated sewage. These results highlight the importance of tracking inputs from these sources in addition to sewer wastewater, particularly for coral. Given the proximity of many of these habitats to human populations, wastewater impacts are additionally compounded by pressure from overfishing and habitat degradation from coastal development, along with climate driven [5] and other anthropogenic

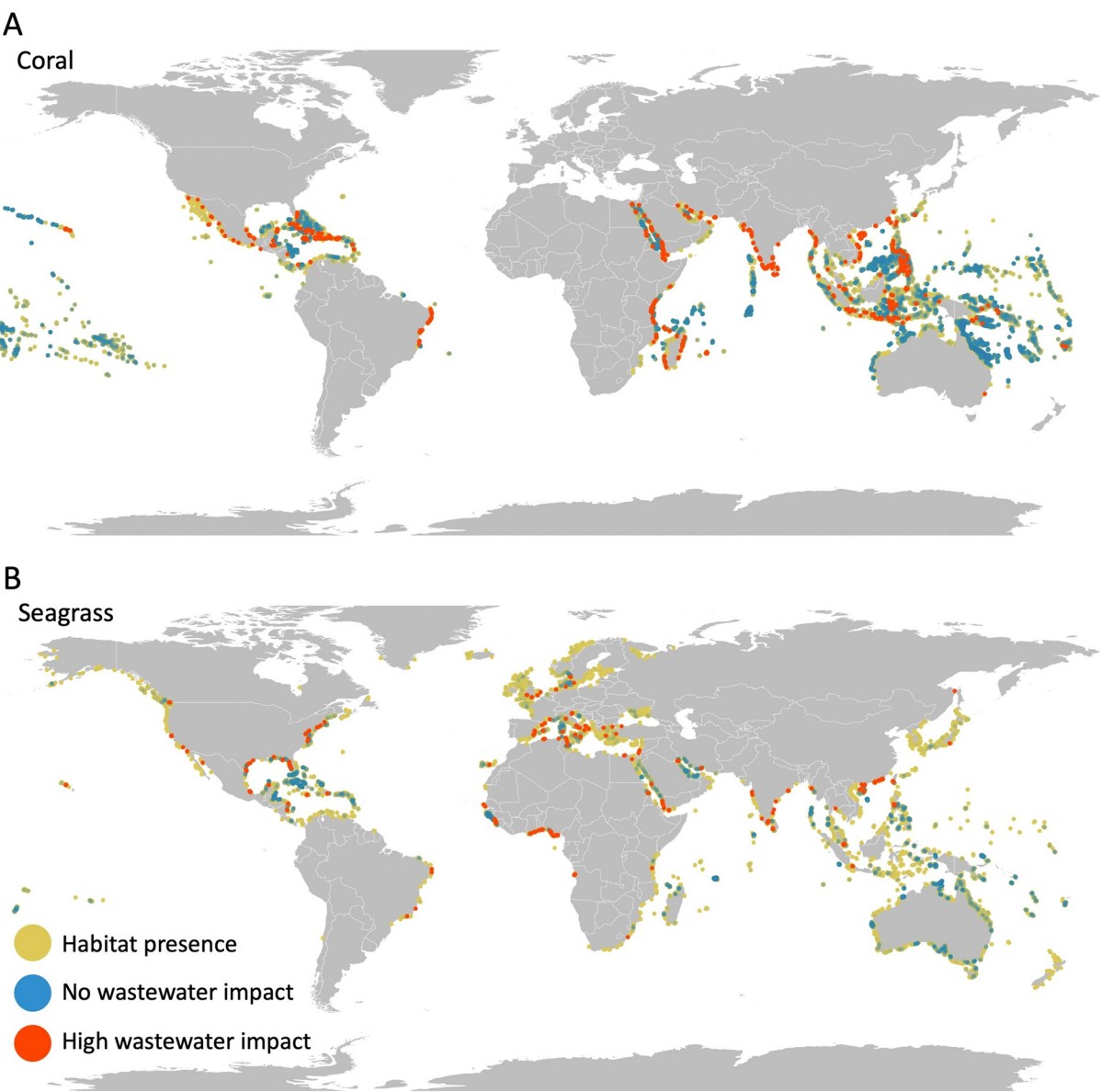

**Fig 3. Expected impact of N on sensitive coastal habitats.** Maps show where A) coral reefs and B) seagrass beds are heavily impacted (raster cells in top 2.5% of exposure; red dots), not impacted (no exposure to wastewater N; dark blue dots), or impacted but not in the top 2.5% (yellow dots). Raster cells are represented as points which visually over-represents the habitat; red is overlaid on top which makes it visually dominant; blue points are transparent and overlaid on green/yellow points such that higher densities of unimpacted areas are brighter blue.

stressors [1]. Addressing wastewater mitigation could help build resilience to other stressors by reducing the overall cumulative impact of human activities [65].

Wastewater carries human pathogens as well as excess N, and therefore poses a risk to both human health and ecological health. To explore this issue, we modeled input into coastal oceans of fecal indicator organisms (FIO) from watersheds and found that just 25 watersheds contribute approximately 51% of FIO into the ocean (S4e Fig in S1 File). These top 25 water-sheds are located on almost every continent, in particular in densely-populated deltas and estu-aries in Southern and Eastern Asia, as well as in Africa (S4e Fig in S1 File). Often N and FIO inputs are correlated, which makes sense given the common source, but in many locations we

find spatial heterogeneity (Fig 4). The 'outliers' when comparing N vs. FIO input (Fig 4) highlight locations where management interventions may have simpler solutions because there is a dominant wastewater concern within the region (either N or pathogens, but not both). The two watersheds with the greatest input values emphasize these differences (Fig 4B): both the Chang Jiang and Brahmaputra (which merges with the Ganges) have very high FIO input, but the Chang Jiang has substantially higher N input while the Brahmaputra has much lower than expected N.

In contrast, for the vast majority of places where N and FIO inputs are strongly correlated, there is a substantial challenge in finding a comprehensive solution. Building and maintaining primary and secondary sewer infrastructure can dramatically reduce FIO levels, but doing so concentrates N with limited capacity to remove it [29, 66], creating large environmental impacts from eutrophication at the location of sewage discharge. Direct and septic inputs into landscapes leverage the capacity of natural ecosystems to take up N but do little to mitigate FIO [52]. This inherent tradeoff between built versus natural infrastructure for wastewater management requires careful attention to the balance between human health and ecological health in management actions. Discussions around green (natural) versus gray (built) infrastructure focus primarily on N removal and thus sidestep these potential tradeoffs, but do not address the daunting (and expensive) task of N removal from built infrastructure. Tertiary wastewater treatment that leverages a range of filtering and chemical processes is needed to meaningfully remove N from wastewater [21], whereas nature can provide this service for free. By mapping N and FIO impacts separately at the watershed scale, our results provide a critical piece of information needed to inform these decisions.

The transboundary nature of watersheds creates a further challenge with the management of wastewater impacts. For the largest 100 watersheds, more than half (N = 53) span national boundaries, and watersheds spanning multiple boundaries (N = 846) account for 54% of the total global watershed area. For example, understanding wastewater input and improving management of wastewater in the Ganges River, which drains Nepal, northeastern India, and parts of Bangladesh, requires sub-national assessment across these three countries. In Europe, the Danube watershed spans 19 countries with different population sizes, cultures, economic development, and levels of wastewater infrastructure and treatment. Even for modest-sized watersheds, many span national boundaries; of the 4,133 watersheds >1,000 km$^2$, 445 span a national boundary. National-level reporting of wastewater inputs (e.g. [21]) makes addressing inputs into coastal waters challenging. Our results provide needed clarity, highlighting FIO and N inputs across all watersheds, as well as by country.

The global model we present here necessarily neglects important details about human communities and ecosystem processes at local scales that would influence actual inputs of N and pathogens from wastewater. Our model does not fully account for differences in land cover, ecosystem processes, sanitation efficiency, and other biogeochemical processes, as we cannot include a process-based model for every watershed (>100,000) on the planet. Most notably, we have applied broad assumptions about N uptake in freshwater systems, even though local-scale vegetation, soil type, slope, precipitation, seasonal dynamics, surface and groundwater flow, and impoundment from dams, among other factors, are known to alter transport and uptake dynamics. For watersheds with high instream retention, this may lead to overestimation of impacts in coastal ecosystems, whereas for others with rapid flush times, we may underestimate coastal impacts [67]. N and FIO inputs may be much higher in many locations than our results suggest because, Furthermore, along with locations of treatment plants and type globally, data on the location of most septic systems and sewer outfalls were modeled rather than observed, and we had no information on septic failure rates (or lack of actually using drain fields) or permitted or accidental sewer overflow events. While this adds uncertainty to

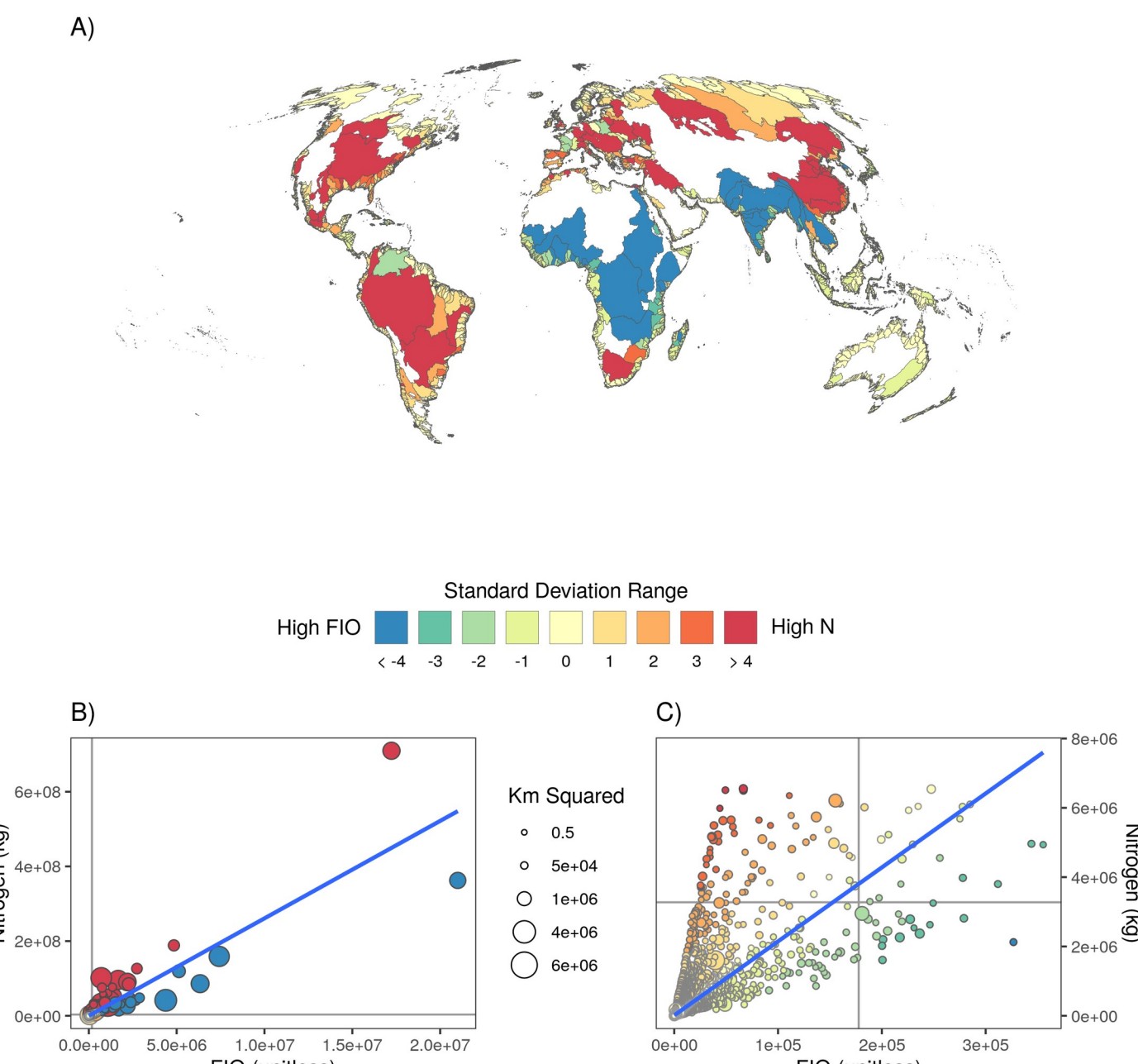

**Fig 4. The relationship between watershed level input of N versus FIO.** Watersheds for which total N exceeds (yellow to red colors) or is well below (green to blue colors) expected levels given predicted nitrogen/pathogen outputs from their correlation. Global map (A) and scatterplots of all watersheds, both full extent (B) and zoomed in (C). Watersheds for which the levels are proportional to expectations are white in the map, inclusive of the nearly 50% of (very small) watersheds for which there is no nitrogen or pathogen inputs. Dots in the scatter plot are scaled to the size of the watershed.

our model, our results may be conservative given high rates of septic tank failure observed in case studies [e.g. 68] and, for treated N, we may overestimate actual N removal rates by treatment plants [28].

We also did not address inputs of phosphorus, despite its role in eutrophication in some ecosystems, or the many other chemicals and pollutants that are in wastewater that can have negative consequences for habitats and species [17]. Like wastewater more broadly, marine

conservation researchers and managers have historically neglected to address the wide variety of wastewater pollutants that impact coastal ecosystems, and while we posit that the locations of impacts of N will be similar to other wastewater pollutants, further research is needed. Finally, we assumed diffusion into coastal waters, given the prohibitive complexity of global coastal advective processes, and we could not account for human migration (in particular through tourism) that would create pulses of wastewater input. Regional applications of our approach would ideally incorporate as many of these details as possible.

Despite these limitations, our approach is the first to produce fine-resolution, comparable data mapping land-based sources and downstream exposure of both pathogens and N in coastal ecosystems. Our global findings are within range of similar studies, but provide a high-resolution understanding of the spatial pattern of relative contribution of wastewater inputs to coastal waters worldwide. Indeed, in terms of impacts to coral reefs, the geographic patterns we map parallel other recent assessments of wastewater inputs to coastal waters [16, 17]. Simulated inputs of diclofenac, an indicator of wastewater, mapped similar locations of wastewater impacts on reef systems [16] in regions like the Tropical Atlantic, Western Indo-Pacific, coastal Western India, and the Central-Indo Pacific (Fig 3), though we show a greater proportion of reef areas impacted. Both models show that in sparsely populated regions like the Eastern Pacific, with a high concentration of reefs, wastewater impacts are minimal. As such, potential interventions at any scale can easily be assessed for environmental and public health consequences, allowing direct comparison of outcomes at the hyper-local to the fully global, and on land and in the ocean.

Our approach uses state-of-the-art, high-resolution data on urbanization and population size, which allows us to identify the relative contribution of N from rural and urban populations worldwide at a much higher resolution than previous studies. The values aggregated to watershed pourpoints and then plumed at high resolution into coastal waters is both novel and of particular interest to marine conservation and public health. The advantage of our model is that is provides a contrast between high spatial resolution demographic drivers and coarse-grained landscape dynamics. We then couple modeled wastewater inputs with high-resolution coastal plume models to capture the two objectives of our project: to map the input of N and pathogens from sewer systems into coastal ecosystems as driven by the heterogeneous demographic patterns worldwide and produce a synthesis to contrast benefits and showcase how these spatial patterns can be used to identify fine-grained trades offs between available solutions relevant for policy makers. Finally, our model and resulting data can be useful for comparing our globally comprehensive results with in situ monitoring and case studies. For instance, stable isotopes can be used to distinguish wastewater N from other anthropogenic sources in coastal waters [68]. Triangulating high resolution global models of N sources and impacts to coastal ecosystems with traditional water quality monitoring and N isotopes analysis, as exemplified by studies from the tributaries to the Chesapeake Bay [68] and Sarasota Bay in the United States [69], can benefit conservation and management with the dual-advantage of combined bottom-up and top-down approaches. Our top-down model can thus be used to identify priority hotspots for in-depth in situ monitoring that can pinpoint the true sources and quantities of pollutants like N.

Wastewater inputs of pathogens and nitrogen into coastal oceans present clear challenges to coastal ecosystems, public health, and economies across the planet. Beyond these direct impacts, our results suggest that wastewater inputs are likely to interact with the plethora of anthropogenic stressors to coastal ecosystems, leading to declining fisheries, habitat loss and degradation, and human health impacts. Climate change is compounding these threats globally [5]. Yet the spatial distribution, overlap, and source of these inputs has remained largely 'off the radar' of most conservation organizations and management agencies. Efforts to address

eutrophication in coastal waters need to give much greater attention to wastewater inputs, and engage directly with public health sectors to balance environmental management actions with human health needs and priorities. Equally, wastewater management to improve public health needs to address the inevitable consequences for ecological health. Without awareness of each issue and coordination across both, coastal ecosystems and the human communities that depend on them will remain imperiled.

## Supporting information

**S1 File. S1 File contains all supporting text, figures and tables.**
(PDF)

## Acknowledgments

The authors would like to thank three anonymous reviewers for their valuable feedback.

## Author Contributions

**Conceptualization:** Benjamin S. Halpern, Juan Carlos Villasenor, Melanie Frazier, Kelly Caylor.

**Data curation:** Cascade Tuholske, Gordon Blasco, Juan Carlos Villasenor, Melanie Frazier.

**Formal analysis:** Cascade Tuholske, Gordon Blasco, Juan Carlos Villasenor, Melanie Frazier.

**Funding acquisition:** Benjamin S. Halpern, Kelly Caylor.

**Investigation:** Cascade Tuholske, Benjamin S. Halpern, Juan Carlos Villasenor, Kelly Caylor.

**Methodology:** Cascade Tuholske, Gordon Blasco, Juan Carlos Villasenor, Melanie Frazier, Kelly Caylor.

**Project administration:** Cascade Tuholske, Benjamin S. Halpern.

**Resources:** Benjamin S. Halpern.

**Software:** Cascade Tuholske, Benjamin S. Halpern, Juan Carlos Villasenor.

**Supervision:** Cascade Tuholske, Benjamin S. Halpern, Kelly Caylor.

**Validation:** Gordon Blasco, Juan Carlos Villasenor.

**Visualization:** Gordon Blasco, Juan Carlos Villasenor, Melanie Frazier.

**Writing – original draft:** Cascade Tuholske, Benjamin S. Halpern, Juan Carlos Villasenor.

**Writing – review & editing:** Cascade Tuholske, Benjamin S. Halpern, Gordon Blasco, Juan Carlos Villasenor, Melanie Frazier, Kelly Caylor.

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
