## [Decision Letter · Decision Letter 0]

31 Aug 2021

PONE-D-21-15046

Mapping global inputs and impacts from of human sewage in coastal ecosystems

PLOS ONE

Dear Dr. Tuholske,

Thank you for submitting your manuscript to PLOS ONE. After careful consideration, we feel that it has merit but does not fully meet PLOS ONE’s publication criteria as it currently stands. Therefore, we invite you to submit a revised version of the manuscript that addresses the points raised during the review process.

We look forward to receiving your revised manuscript.

Kind regards,

Bijeesh Kozhikkodan Veettil

Academic Editor

PLOS ONE

1. Please ensure that your manuscript meets PLOS ONE's style requirements, including those for file naming. The PLOS ONE style templates can be found at https://journals.plos.org/plosone/s/file?id=wjVg/PLOSOne_formatting_sample_main_body.pdf and https://journals.plos.org/plosone/s/file?id=ba62/PLOSOne_formatting_sample_title_authors_affiliations.pdf.

4. We note that Figures 1,3 and 4 in your submission contain [map/satellite] images which may be copyrighted. All PLOS content is published under the Creative Commons Attribution License (CC BY 4.0), which means that the manuscript, images, and Supporting Information files will be freely available online, and any third party is permitted to access, download, copy, distribute, and use these materials in any way, even commercially, with proper attribution. For these reasons, we cannot publish previously copyrighted maps or satellite images created using proprietary data, such as Google software (Google Maps, Street View, and Earth). For more information, see our copyright guidelines: http://journals.plos.org/plosone/s/licenses-and-copyright.

 a. You may seek permission from the original copyright holder of Figures 1,3 and 4 to publish the content specifically under the CC BY 4.0 license. 

Additional Editor Comments (if provided):

Reviewers' comments:

Reviewer's Responses to Questions

**Comments to the Author**

1. Is the manuscript technically sound, and do the data support the conclusions?

Reviewer #1: Yes

Reviewer #2: Yes

Reviewer #3: Yes

2. Has the statistical analysis been performed appropriately and rigorously? 

Reviewer #1: Yes

Reviewer #2: Yes

Reviewer #3: Yes

3. Have the authors made all data underlying the findings in their manuscript fully available?

Reviewer #1: Yes

Reviewer #2: Yes

Reviewer #3: Yes

4. Is the manuscript presented in an intelligible fashion and written in standard English?

Reviewer #1: Yes

Reviewer #2: Yes

Reviewer #3: Yes

5. Review Comments to the Author

Reviewer #1: This work by Tuholske et al. represents a major step forward in our understanding of global threats to coastal ecosystems and people. Current theory in marine ecology and conservation is that pathogen and nutrient threats emanating from wastewater, although intense at times and locations, are very limited in space and only affect a small portion of the worlds coast and coastal ecosystems. Indeed, the prevailing dogma throughout marine ecology and even in the environmental management/conservation space is that the massive nutrient threat that now imperils coastal ecosystems worldwide emanates from agriculturally-derived nutrients. The present work uses data-informed models to overturn this paradigm and show that nearly half of the nitrogen dumping into coastal ecosystems comes from wastewater and that these nutrients threaten well over half of the worlds seagrasses and coral reefs.

This is a very important paper and more than meets the criteria for publication in PLOS ONE. This work will be widely read and I will certainly feature this work prominently in my outreach, education, and training activities.

Everyone in marine ecology and conservation knows that nutrients threaten our coasts. What we didn’t know that is revealed in this paper is that half of those nutrients come from wastewater. As a community, we had assumed that the primary, overwhelming source was from agriculture and, accordingly, have spent most of our efforts abating and controlling for agricultural run off. Now we must shift quickly to also focus on wastewater treatment, management, and reduce wastewater flows into coastal and riverine systems.

The writing is very good, and the model is sound. Of course, like many models, this work is limited by many assumptions but the authors feature those caveats and limitations in the discussion prominently. Still, the findings are robust represent a major realignment of the threat paradigm in marine ecology and conservation.

Some minor/moderate, but essential, editing will improve the readability and science behind of the ms.

1. Very importantly, these results are generated by modeling, so are not actual N load data. There is no way around this given the data limitations in the world. To accurately reflect this in the ms, the authors should carefully and consistently edit the ms to convey that their data were generated from a model. For instance using the word “predicts” not “shows” is more appropriate. This is done well in the discussion but not in the abstract.

2. As it is a model, the authors should talk about next steps that need to be taken to test the model and confront it with real data. This should be done where data already exist of course. In fact, could the authors test their models in well-studied systems such as the Chesapeake or Tampa Bay where already published studies use isotopes to estimate the relative contribution to nitrogen loading in these waters by wastewater vs. agricultural sources?

3. A recent, complementary global modeling paper (Wear et al 2021 Bio. Con.) https://www.sciencedirect.com/science/article/pii/S0006320721000628 that estimated the reach of wastewater pollution in coastal and some river systems similarly estimated the severity and extent of some coastal habitats (coral reefs, salt marshes, and fish-rich rivers) that are being impacted by wastewater. A paragraph or few lines in the discussion comparing and contrasting Wear et al.’s results with the results from the current paper is appropriate. The current ms importantly is very unique in that it estimates amount of N coming into the system and its sources.

4. This work focuses on 2 components of wastewater – N and pathogens. Currently that is mostly what the rest of the world thinks are the primary components of wastewater. But it’s actually a combination of many dangerous components, including endocrine disruptors, microplastics, heavy metals, pharmaceuticals, etc. Please see Wear and Vega Thurber 2015 in Annals of the NY Academy of Sciences for a review on the common components and their impacts on coral reef organisms. Briefly talking about this in more detail in the discussion is appropriate and important – especially given a history of this being poorly understood and a growing body of research examining these impacts on specific species and systems (e.g., recent studies on trout addicted to methamphetimines-https://www.nature.com/articles/d41586-021-01846-7).

5. Citation #10 is only for coral reefs, but mangroves and marshes are mentioned here. Please add in citations that show impacts of nutrients on those systems. C. Lovelock has an outstanding paper on this for mangroves and many other related papers on impacts of eutrophication (https://scholar.google.com/citations?view_op=view_citation&hl=en&user=PZfbYkAAAAAJ&cstart=200&pagesize=100&sortby=pubdate&citation_for_view=PZfbYkAAAAAJ:Wp0gIr-vW9MC). For salt marshes both M. Bertness (Bertness et al. 2003 PNAS) and L. Deegan have separately published on this topic. https://www.nature.com/articles/nature11533

6. For the discussion about interactions between local human threats and climate change, I would suggest updating those discussions with more recent work from C. Harley’s review, and the most review on this topic by He and Silliman Current Biology 2019.

7. For this sentence: “While research has assessed impacts of sewered wastewater on coastal ecosystems and community health, we lack a comprehensive, fine-resolution, global assessment of the inputs and impacts human sewage that captures both pathogens and nutrient flows to coastal waters and impacts on coastal ecosystems” I would note that very little research has actually assessed the impacts – you will note in the Wear & Vega Thurber 2015 paper that very few papers documented these impacts in a comprehensive way in coral reef systems (negative impacts are broadly assumed but not well documented or tested) (https://ir.library.oregonstate.edu/concern/articles/8910jz63t) . There is a clear lack of research in this space. However, this paper will do much to spotlight the need to better understand these impacts.

8. Also, please rewrite this sentence in the second paragraph of intro to improve clarity. “In contrast to agricultural inputs, globally comprehensive impacts from human sewage on coastal waters have received much less attention, much less with high spatial resolution data.”

9. In the model used, average removal N rates for the various treatments of wastewater are used. What if the authors also generated high and low estimates in their models based on high and low efficiency. I believe that the current model results overestimate how good treatment works (e.g., 80% in septic – I have seen studies that showed about 35% removal- Ritter and Eastburn 1988) and thus the reality is that more N in coastal waters is derived from wastewater than even this model predicts. Is there a way to talk about or generate data on the variance in this model?

10. Linked to this, given all the assumptions that were made, is the current model more likely to over or underestimate the real contribution of wastewater to coastal N? It seems to underestimate it.

Reviewer #2: The abstract is suggested to be re-written summarizing short introduction, problem statement, methodology, major results and final conclusion and recommendation

The abbrivations in the abstract and throughout manuscript should put in full name for first time

The novelty of this study needs to be clearly highlighted in terms of the environmental advantages.

Reviewer #3: General

1. This is an important contribution, and my comments are mostly concerned with grammar.

2. Define “pourpoint” upon first use (i.e., the point at which water flows from the watershed to the coastal ocean).

3. Replace “environmental health” with “ecological health” throughout.

Specific

1. Please reference the following along with reference #2:

Malone, T.C. and A. Newton. 2020. The globalization of cultural eutrophication in the coastal ocean: Causes and consequences. Front. Mar. Sci. 7:670. doi: 10.3389/fmars.2020.00670

2. p.3, line 1: This should read “…risk from wastewater has largely occurred in isolation [26].”

3. p.3, line 9: This should read “…focus primarily on the largest watersheds…”

4. p. 3, line 15: “Primarily” is repeated twice. Delete one.

5. p. 3, line 20: This should read “…can provide context for specific details, such as how fecal…”

6. p. 3, line 22: This should read “…[39], e.g., global pathogen wastewater…”

7. p. 3, line 24: What are “sub-national policy recommendations”?

8. p. 11, line 17: “…alone with climate change and other anthropogenic stressors [1].” Replace “climate change” with “climate driven stressors (e.g., ocean warming, acidification and sea level rise).

9. p. 13, line 3: This should read “…ecosystems to take up N but do…”

10. p. 14, line 3: Replace “known” with “observed”.

11. p. 15, line 4: Delete “vast majority of the”.

6. PLOS authors have the option to publish the peer review history of their article (what does this mean?). If published, this will include your full peer review and any attached files.

Reviewer #1: No

Reviewer #2: No

Reviewer #3: **Yes: **Thomas C. Malone

---

## [Author Response · Author response to Decision Letter 0]

27 Sep 2021

Response to Reviewers’ Comments

"Mapping global inputs and impacts from of human sewage in coastal ecosystems"

PONE-D-21-15046

Content

1. Editor

2. Reviewer 1

3. Reviewer 2

4. Reviewer 3

Comments and questions from the editor and reviewers listed below in black. Our responses to each comment and question are in blue. 

1. Please ensure that your manuscript meets PLOS ONE's style requirements, including those for file naming. The PLOS ONE style templates can be found at https://journals.plos.org/plosone/s/file?id=wjVg/PLOSOne_formatting_sample_main_body.pdf and https://journals.plos.org/plosone/s/file?id=ba62/PLOSOne_formatting_sample_title_authors_affiliations.pdf.

Thank you for catching this error. We will update both sections accordingly. 

We have prepared a DOI for the dataset and added it to the manuscript, along with a link to the Github repository for the code required to reproduce our analysis. 

4. We note that Figures 1, 3 and 4 in your submission contain [map/satellite] images which may be copyrighted. All PLOS content is published under the Creative Commons Attribution License (CC BY 4.0), which means that the manuscript, images, and Supporting Information files will be freely available online, and any third party is permitted to access, download, copy, distribute, and use these materials in any way, even commercially, with proper attribution. For these reasons, we cannot publish previously copyrighted maps or satellite images created using proprietary data, such as Google software (Google Maps, Street View, and Earth). For more information, see our copyright guidelines: http://journals.plos.org/plosone/s/licenses-and-copyright.

 a. You may seek permission from the original copyright holder of Figures 1,3 and 4 to publish the content specifically under the CC BY 4.0 license. 

 n the figure caption of the copyrighted figure, please include the following text: “Reprinted from [ref] under a CC BY license, with permission from [name of publisher], original copyright [original copyright year].”

National boundaries in Figures 1 and 3, as well as the Supplement, are from the GADM data catalogue and Natural Earth. These data are publicly available for use of academic publishing without prior permission. 

The GADAM licenses (available here and uploaded as requested: https://gadm.org/license.html) states: “The data are freely available for academic use and other non-commercial use. Redistribution, or commercial use, is not allowed without prior permission. Using the data to create maps for academic publishing is allowed.”

Watershed boundaries in Fig. 4 were produced by the authors. 

Additional Editor Comments (if provided):

Reviewers' comments:

Reviewer's Responses to Questions

Comments to the Author

1. Is the manuscript technically sound, and do the data support the conclusions?

Reviewer #1: Yes

Reviewer #2: Yes

Reviewer #3: Yes

2. Has the statistical analysis been performed appropriately and rigorously?

Reviewer #1: Yes

Reviewer #2: Yes

Reviewer #3: Yes

3. Have the authors made all data underlying the findings in their manuscript fully available?

Reviewer #1: Yes

Reviewer #2: Yes

Reviewer #3: Yes

4. Is the manuscript presented in an intelligible fashion and written in standard English?

Reviewer #1: Yes

Reviewer #2: Yes

Reviewer #3: Yes

5. Review Comments to the Author

Reviewer #1: This work by Tuholske et al. represents a major step forward in our understanding of global threats to coastal ecosystems and people. Current theory in marine ecology and conservation is that pathogen and nutrient threats emanating from wastewater, although intense at times and locations, are very limited in space and only affect a small portion of the worlds coast and coastal ecosystems. Indeed, the prevailing dogma throughout marine ecology and even in the environmental management/conservation space is that the massive nutrient threat that now imperils coastal ecosystems worldwide emanates from agriculturally-derived nutrients. The present work uses data-informed models to overturn this paradigm and show that nearly half of the nitrogen dumping into coastal ecosystems comes from wastewater and that these nutrients threaten well over half of the worlds seagrasses and coral reefs.

This is a very important paper and more than meets the criteria for publication in PLOS ONE. This work will be widely read and I will certainly feature this work prominently in my outreach, education, and training activities.

Everyone in marine ecology and conservation knows that nutrients threaten our coasts. What we didn’t know that is revealed in this paper is that half of those nutrients come from wastewater. As a community, we had assumed that the primary, overwhelming source was from agriculture and, accordingly, have spent most of our efforts abating and controlling for agricultural run off. Now we must shift quickly to also focus on wastewater treatment, management, and reduce wastewater flows into coastal and riverine systems.

The writing is very good, and the model is sound. Of course, like many models, this work is limited by many assumptions but the authors feature those caveats and limitations in the discussion prominently. Still, the findings are robust represent a major realignment of the threat paradigm in marine ecology and conservation.

Some minor/moderate, but essential, editing will improve the readability and science behind of the ms.

1. Very importantly, these results are generated by modeling, so are not actual N load data. There is no way around this given the data limitations in the world. To accurately reflect this in the ms, the authors should carefully and consistently edit the ms to convey that their data were generated from a model. For instance using the word “predicts” not “shows” is more appropriate. This is done well in the discussion but not in the abstract.

We thank the review for this helpful critique. We have updated the abstract, as well as the methods and results/discussion sections of the manuscript, to emphasize that our findings are model-based. 

2. As it is a model, the authors should talk about next steps that need to be taken to test the model and confront it with real data. This should be done where data already exist of course. In fact, could the authors test their models in well-studied systems such as the Chesapeake or Tampa Bay where already published studies use isotopes to estimate the relative contribution to nitrogen loading in these waters by wastewater vs. agricultural sources?

This is an excellent suggestion. While we do not have a comprehensive inventory of isotope case studies, we have added this to our discussion [Pg. 15, ¶ 2] to tee up next steps with our model. We have added the following references as well:

Fertig, B., Carruthers, T. J., & Dennison, W. C. (2014). Oyster δ15N as a bioindicator of potential wastewater and poultry farming impacts and degraded water quality in a subestuary of Chesapeake Bay. Journal of Coastal Research, 30(5), 881-892. [maybe]

3. A recent, complementary global modeling paper (Wear et al 2021 Bio. Con.) https://www.sciencedirect.com/science/article/pii/S0006320721000628 that estimated the reach of wastewater pollution in coastal and some river systems similarly estimated the severity and extent of some coastal habitats (coral reefs, salt marshes, and fish-rich rivers) that are being impacted by wastewater. A paragraph or few lines in the discussion comparing and contrasting Wear et al.’s results with the results from the current paper is appropriate. The current ms importantly is very unique in that it estimates amount of N coming into the system and its sources.

Thank you for bringing Wear et al. 2021 to our attention, which looks at the impact of wastewater chemicals on coral and saltmarsh habitats as well as freshwater fisheries. We compared the findings for coral reefs, a habitat included in both projects, and the spatial footprint of high and low impact areas appear very similar. We now discuss this comparison in the paper [Pg. 14-15]

4. This work focuses on 2 components of wastewater – N and pathogens. Currently that is mostly what the rest of the world thinks are the primary components of wastewater. But it’s actually a combination of many dangerous components, including endocrine disruptors, microplastics, heavy metals, pharmaceuticals, etc. Please see Wear and Vega Thurber 2015 in Annals of the NY Academy of Sciences for a review on the common components and their impacts on coral reef organisms. Briefly talking about this in more detail in the discussion is appropriate and important – especially given a history of this being poorly understood and a growing body of research examining these impacts on specific species and systems (e.g., recent studies on trout addicted to methamphetimines-https://www.nature.com/articles/d41586-021-01846-7).

We agree with the reviewer and have added more detail in the discussion and introduction about how our findings only offer insights into quantifying N and Pathogen impacts. 

5. Citation #10 is only for coral reefs, but mangroves and marshes are mentioned here. Please add in citations that show impacts of nutrients on those systems. C. Lovelock has an outstanding paper on this for mangroves and many other related papers on impacts of eutrophication (https://scholar.google.com/citations?view_op=view_citation&hl=en&user=PZfbYkAAAAAJ&cstart=200&pagesize=100&sortby=pubdate&citation_for_view=PZfbYkAAAAAJ:Wp0gIr-vW9MC). For salt marshes both M. Bertness (Bertness et al. 2003 PNAS) and L. Deegan have separately published on this topic. https://www.nature.com/articles/nature11533

We have updated the introduction with full citations for each ecosystem impact accordingly. 

6. For the discussion about interactions between local human threats and climate change, I would suggest updating those discussions with more recent work from C. Harley’s review, and the most review on this topic by He and Silliman Current Biology 2019.

We have updated the discussion on interactions between climate change and humans relevant to coastal ecosystems with He and Silliman 2019. We were unsure of the C. Harley’s “recent work”, but we added the widely-cited and relevant:

Harley, C. D., Randall Hughes, A., Hultgren, K. M., Miner, B. G., Sorte, C. J., Thornber, C. S., ... & Williams, S. L. (2006). The impacts of climate change in coastal marine systems. Ecology letters, 9(2), 228-241.

7. For this sentence: “While research has assessed impacts of sewered wastewater on coastal ecosystems and community health, we lack a comprehensive, fine-resolution, global assessment of the inputs and impacts human sewage that captures both pathogens and nutrient flows to coastal waters and impacts on coastal ecosystems” I would note that very little research has actually assessed the impacts – you will note in the Wear & Vega Thurber 2015 paper that very few papers documented these impacts in a comprehensive way in coral reef systems (negative impacts are broadly assumed but not well documented or tested) (https://ir.library.oregonstate.edu/concern/articles/8910jz63t) . There is a clear lack of research in this space. However, this paper will do much to spotlight the need to better understand these impacts.

This is a useful critique and we have updated the abstract as follows:

“Few studies have assessed impacts of human wastewater on coastal ecosystems and community health. As such, we lack a comprehensive, fine-resolution, global assessment of human sewage inputs that capture both pathogens and nutrient flows to coastal waters and the potential impacts on coastal ecosystems.”

8. Also, please rewrite this sentence in the second paragraph of intro to improve clarity. “In contrast to agricultural inputs, globally comprehensive impacts from human sewage on coastal waters have received much less attention, much less with high spatial resolution data.”

We have updated this sentence: 

“In contrast to agricultural inputs, few high-resolution, globally-comprehensive studies have measured and mapped the impacts from human sewage on coastal ecosystems (Wear et al. 2021; Wear and Thurber 2015).”

9. In the model used, average removal N rates for the various treatments of wastewater are used. What if the authors also generated high and low estimates in their models based on high and low efficiency. I believe that the current model results overestimate how good treatment works (e.g., 80% in septic – I have seen studies that showed about 35% removal- Ritter and Eastburn 1988) and thus the reality is that more N in coastal waters is derived from wastewater than even this model predicts. Is there a way to talk about or generate data on the variance in this model?

We appreciate this concern and agree that our model necessarily is conservative, especially for populations living in close proximity to coastal waters and fluvial systems without significant N up-take. We have added language in the discussion [Pg. 13 – 14] to highlight this in more detail. Furthermore, all our code and model outputs are publicly available for researchers and managers to assess the sensitivity of the removal parameter, as well as any other input data. 

10. Linked to this, given all the assumptions that were made, is the current model more likely to over or underestimate the real contribution of wastewater to coastal N? It seems to underestimate it.

 This is a challenging question that depends on fine-resolution biophysical processes that we cannot account for across all watersheds worldwide. For instance, our model does not account for N (or FIO) inputs to fluvial systems from groundwater, which would lead us to underestimate inputs. On the other hand, we do not measure different flushing times and seasonality of flows that would produce heterogeneous spatiotemporal differences in stream N retention. On the human side, as you noted and we cite in the paper [see Bouwman et al. 2005], N/FIO removal rates from different wastewater treatment systems vary greatly by location. Given that many wastewater plants do not remove N our estimates may again under estimate N inputs in many locations.

 As such, we emphasize throughout that our model's objective is to map relative inputs and document extremes across all watersheds globally to identify hotspots and provide managers with information regarding tradeoffs and benefits of interventions. While the model is a based on underlying fusion of datasets that themselves lack uncertainty (e.g. JRC rates of wastewater treatment types), our model produces a representation of the true distribution of wastewater impacts worldwide. We have added further details in the discussion to address this point [Pg. 13 – 14].

Bouwman AF, Van Drecht G, Knoop JM, Beusen AHW, Meinardi CR. Exploring changes in river nitrogen export to the world’s oceans. Global Biogeochem Cycles. 2005;19: 704.

Reviewer #2: The abstract is suggested to be re-written summarizing short introduction, problem statement, methodology, major results and final conclusion and recommendation

We have written our abstract as described and defer to the editorial policies of the journal if we should reformat our abstract to have subheadings. 

The abbrivations in the abstract and throughout manuscript should put in full name for first time

We have copy-edited the abstract and manuscript and written all abbreviations in full at the first use.

The novelty of this study needs to be clearly highlighted in terms of the environmental advantages.

We appreciate the reviewer’s comment and have added more detailed language to the abstract highlighting how our analysis helps researchers and practitioners understand the environmental impact of human wastewater and provides a new high-resolution lens in which to focus mitigation efforts. In short, by mapping wastewater inputs, and contrasting when nitrogen and pathogens inputs overlap and diverge, we can determine where habitats and species will be most vulnerable to the pressures caused by wastewater, as well as examine areas of human health concerns. This aids with interdisciplinary collaborations to solve the complex challenges wastewater presents for coastal ecosystems and human health. We direct the reviewer to these sections [e.g. Pg. 14 - 15] where we showcase the novelty and utility of our analysis. 

Reviewer #3: General

1. This is an important contribution, and my comments are mostly concerned with grammar.

2. Define “pourpoint” upon first use (i.e., the point at which water flows from the watershed to the coastal ocean).

This is done on Pg. 4, ¶ 2.

3. Replace “environmental health” with “ecological health” throughout.

Thank you - we have updated the working throughout the manuscript. 

Specific

The authors thank the reviewer for the diligent copy editing. It is much appreciated! 

1. Please reference the following along with reference #2:

Malone, T.C. and A. Newton. 2020. The globalization of cultural eutrophication in the coastal ocean: Causes and consequences. Front. Mar. Sci. 7:670. doi: 10.3389/fmars.2020.00670

Thank you - have added this reference. 

2. p.3, line 1: This should read “…risk from wastewater has largely occurred in isolation [26].”

We have made the correction.

3. p.3, line 9: This should read “…focus primarily on the largest watersheds…”

We have made the correction.

4. p. 3, line 15: “Primarily” is repeated twice. Delete one.

We have made the correction.

5. p. 3, line 20: This should read “…can provide context for specific details, such as how fecal…”

We have made the correction.

6. p. 3, line 22: This should read “…[39], e.g., global pathogen wastewater…”

We have made the correction.

7. p. 3, line 24: What are “sub-national policy recommendations”?

We corrected this to ‘regional and local’.

8. p. 11, line 17: “…alone with climate change and other anthropogenic stressors [1].” Replace “climate change” with “climate driven stressors (e.g., ocean warming, acidification and sea level rise).

We have made the correction.

9. p. 13, line 3: This should read “…ecosystems to take up N but do…”

We have made the correction.

10. p. 14, line 3: Replace “known” with “observed”.

We have made the correction.

11. p. 15, line 4: Delete “vast majority of the”.

We have made the correction.

6. PLOS authors have the option to publish the peer review history of their article (what does this mean?). If published, this will include your full peer review and any attached files.

Do you want your identity to be public for this peer review? For information about this choice, including consent withdrawal, please see our Privacy Policy.

Reviewer #1: No

Reviewer #2: No

Reviewer #3: Yes: Thomas C. Malone

---

## [Editor Report · Decision Letter 1]

8 Oct 2021

Mapping global inputs and impacts from of human sewage in coastal ecosystems

PONE-D-21-15046R1

Dear Dr. Tuholske,

We’re pleased to inform you that your manuscript has been judged scientifically suitable for publication and will be formally accepted for publication once it meets all outstanding technical requirements.

Kind regards,

Bijeesh Kozhikkodan Veettil

Academic Editor

PLOS ONE
---

## [Editor Report · Acceptance letter]

15 Oct 2021

PONE-D-21-15046R1 

Mapping global inputs and impacts from of human sewage in coastal ecosystems 

Dear Dr. Tuholske:

I'm pleased to inform you that your manuscript has been deemed suitable for publication in PLOS ONE. Congratulations! Your manuscript is now with our production department. 

Kind regards, 

on behalf of

Dr. Bijeesh Kozhikkodan Veettil 

Academic Editor

PLOS ONE